# New Insights into Immunotherapy Strategies for Treating Autoimmune Diabetes

**DOI:** 10.3390/ijms20194789

**Published:** 2019-09-26

**Authors:** Miriam Cabello-Olmo, Miriam Araña, Ilian Radichev, Paul Smith, Eduardo Huarte, Miguel Barajas

**Affiliations:** 1Biochemistry Area, Health Science Department, Faculty of Health Sciences, Public University of Navarra, 31008 Pamplona, Spain; miriamcabelloolmo@gmail.com (M.C.-O.); miriam.arana@unavarra.es (M.A.); 2Diabetes Research Group at Sanford Research, Sioux Falls, SD 57104, USA; Ilian.Radichev@SanfordHealth.org; 3Incyte Corporation, Wilmington, DE 19803, USA; PSmith@incyte.com

**Keywords:** diabetes, autoimmunity, insulin, immunotherapy and clinical trials

## Abstract

Type 1 diabetes mellitus (T1D) is an autoimmune illness that affects millions of patients worldwide. The main characteristic of this disease is the destruction of pancreatic insulin-producing beta cells that occurs due to the aberrant activation of different immune effector cells. Currently, T1D is treated by lifelong administration of novel versions of insulin that have been developed recently; however, new approaches that could address the underlying mechanisms responsible for beta cell destruction have been extensively investigated. The strategies based on immunotherapies have recently been incorporated into a panel of existing treatments for T1D, in order to block T-cell responses against beta cell antigens that are very common during the onset and development of T1D. However, a complete preservation of beta cell mass as well as insulin independency is still elusive. As a result, there is no existing T1D targeted immunotherapy able to replace standard insulin administration. Presently, a number of novel therapy strategies are pursuing the goals of beta cell protection and normoglycemia. In the present review we explore the current state of immunotherapy in T1D by highlighting the most important studies in this field, and envision novel strategies that could be used to treat T1D in the future.

## 1. Introduction

Type 1 diabetes mellitus (T1D) is a pathology emerging from the selective elimination of pancreatic insulin-producing beta cells mediated by an autoimmune defect. Consequently, the main characteristic of this disease that occurs in its advanced stages is hyperglycemia. This form of diabetes accounts for approximately 5–10% of all diabetic patients.

The prevalence of this pathology indicates that more than 500,000 children suffer from type 1 diabetes worldwide, mostly in North America and Europe [1]. However, the epidemiologic studies suggest that the incidence of T1D has increased markedly in recent years [2]. In 2017, the International Diabetes Federation (IDF; https://diabetesatlas.org) declared 132,600 newly diagnosed T1D cases worldwide.

### 1.1. Genetics of T1D

One of the main characteristics of T1D is the loss of beta cell tolerance, a process that involves different factors [3,4] including genetic associations with human leukocyte antigen haplotypes (HLA) and several beta cell-specific genes [5].

T1D is described as an inflammatory disease in which the infiltration of the pancreatic islets with a number of immune cell types (CD4^+^ and CD8^+^ T-cells, macrophages, dendritic cells (DC), and B cells) play a significant role [6]. The progression of islet infiltrates promotes beta cell elimination that ultimately results in the onset of diabetes. 

While having some benefits, the transplantation of pancreas or pancreatic islets (Edmonton Protocol) [7] have had limited success due to the insufficient number of donors and the reactivation of the autoimmunity status despite immunosuppression protocols. Additionally, pancreas transplants have been demonstrated to be only partially successful [8].

### 1.2. Immunological Mechanisms Involved in T1D Pathogenesis

The progression of T1D can be divided into three critical stages [9,10]. At the “first stage”, which may take place through a long period of time, individuals develop beta cell autoimmunity, identified by serum autoantibodies. The most frequent autoantibodies in T1D patients are those against GAD (GAD65), the tyrosine phosphatases IA-2 and IA-2β, zinc transporter 8 (ZnT8), and insulin [11]. Those epitopes can induce the activation of CD4^+^ and CD8^+^ T-cells, which are the main mediators of beta cell destruction. 

The presence of diabetes autoantibodies plays an important role in the identification of preclinical stages of T1D. The TrialNet TN01 has analyzed the importance of the autoantibodies markers for the detection of diabetes [12]. Five percent of the people screened through this study were found to present blood autoantibody. This study also determined that 95% of patients that progress to symptomatic T1D were autoantibody positive by the age of 5 years [13].

The identification of autoantibodies in the TEDDY (The Environmental Determinants of Diabetes in the Young) study showed a peak between 2 and 9 years of age [14]. Individuals that demonstrate the presence of at least two different autoantibodies have a significant chance of developing T1D [12,15].

Additionally, different HLA haplotypes were identified to be either protective or predisposing to diabetes development [16]. When autoreactive CD4^+^ and CD8^+^ T-cells begin to extinguish beta cells, the insulin levels start to decrease, which initiate the “second stage” of the T1D. At this stage, the main strategy for T1D treatment would be to suppress beta cell autoimmunity along with protection of the remaining beta cell mass. Different studies have demonstrated that at the time of diagnosis, which overlap with the second stage, there are still residual beta cells present (Clinical trial NCT01030861) [17]. Administration of immunosuppressive drugs in children with new onset of T1D can delay or reverse diabetes progression; however, immunosuppression can also result in organ toxicity. The diabetes progression resumes once the treatment is withdrawn [18].

The “third stage” of T1D occurs in long-term patients. At this stage, the main objective is to ensure the functionality of the remaining beta cell. The studies have shown that following the disease onset there is a considerable reduction in C-peptide levels, a short polypeptide that connects insulin’s chains in the proinsulin molecule and can be used as a surrogate of how much insulin is produced (Clinical trial NCT01030861) [17]. The maintenance of a high beta cell number could help in the control of hyperglycemia as well as to reduce the comorbidities of the disease.

It has been demonstrated that CD8^+^ and CD4^+^ T-cells, macrophages, and B cells are present in human cadaveric T1D pancreata [12,19]. However, the lack of insulitis in some T1D cadaveric pancreata samples underlines the heterogeneity of the disease [20] which could be one of the reasons why immunotherapies have not been fully effective in T1D patients.

Different immunotherapies have been proposed for all three stages of T1D. One approach involves the manipulation of the immune response, by using antibodies that target specific immune mediators. Another approach takes advantage of beta cell antigen-specific treatments. Interestingly, a treatment based on oral insulin administration demonstrated a delay in the diabetes onset in Non-Obese Diabetic (NOD) mice [21], an animal model which has been heavily used to study the progression and pathogenesis of T1D, and which we will describe in the next section. 

### 1.3. Animal Models of T1D

Two different animal models have mostly been used in the field of T1D research: The NOD mouse and Bio-breeding (BB) rat. Both models exhibit the main symptoms of diabetes: Glycosuria, polyuria, weight loss, and islet of Langerhans-lymphocytic infiltration [22,23]. However, due to the implication of the T-cell compartment in the pathogenesis of T1D, the NOD model has been preferably used for the study of the diabetogenic T-cells development [24]. NOD mice show similar characteristics to human diabetes, summarized in Table 1.

NOD mice were originally generated in the Cataract Shionogi (CTS) strain [22]. Cell infiltration in the pancreas of NOD mice can be observed at as early as 3 weeks of age. This process includes the recruitment of different innate immune cells into the islets of Langerhans including macrophages and neutrophils, prior to the infiltration of the lymphocytes [25,26]. Although the presence of autoreactive T-cells is initially low, their numbers gradually increase, due to the recognition of certain diabetes-specific autoantigens and become activated, initiating the elimination of insulin-producing beta cells. Despite the focus of T1D research on T-cell-mediated beta-cell destruction, there are studies showing that B cells also play an important role in the diabetes onset [27]. 

The NOD mouse model has provided valuable information regarding the role of the immune cells in diabetes development. Furthermore, NOD mice have provided a unique research tool in order to explore immunotherapy treatments (i.e., CTLA4-Ig, anti-CD40 antibodies, and IL-4 or IL-10 treatment), as has been exhaustively reviewed by Shoda et al. [28]. However, most of the immune-interventions that have shown promise in the NOD mouse model failed to demonstrate similar impact on human disease. For this reason, the attempts to humanize NOD mice [29] might facilitate the research that would eventually translate into successful immunotherapy clinical trials.

Additionally, some external factors also play an important role in T1D development. The studies in monozygotic twins have demonstrated a lack of concordance suggesting the importance of environmental factors in the T1D progress. Many of those factors have been involved in modifying diabetes susceptibility in NOD mice, including changes in the gut microbiota [30,31,32]. The interaction of innate immune components with the gut microbiota represents a hot topic in the field of T1D research. We will deepen this aspect in Section 6.2.

Apart from the previously mentioned mouse models, another useful model is the DO11.10×RIPmOVA (DORmO) mouse model, where RIPmOva animals (mice that express membrane-bound OVA in thymus and pancreas) are crossed with DO11 animals expressing an OVA-specific MHC-II TCR. Somehow surprisingly, these double-transgenic animals generate large numbers of islet specific functional T_reg_ cells (see Section 2.3), but spontaneously develop T1D by week 20. Therefore, the DORmO model is uniquely suited to study T_reg_ role in T1D initiation/progression [33,34].

### 1.4. Present State of T1D Immunotherapy

Current strategies for T1D immunotherapies could be classified as antigen-independent and antigen-dependent. Antigen-independent (non-antigen specific) interventions include: drugs that induce immunosuppression, antibody-based therapies that allow the depletion of polyclonal B or T cells [35], cytokine-based strategies [36], and the increase of tolerogenic DC [37], and polyclonal T_reg_ cell numbers [34].

Antigen-dependent (antigen-specific) strategies involve the use of beta cell-derived autoantigen-based vaccines, adoptive transfer strategies and specific abrogation of autoreactive T-cell clone by targeting antigen presentation mechanisms [38,39].

The combination of different interventions based in immunotherapy treatments is considered the most effective strategy due to the complexity of T1D [40].

In the next sections, we will analyze the strategies of immunotherapy that are currently used for treatment and prevention of T1D (summarized in Table 2 and Figure 1).

## 2. Antigen-Independent Strategies

### 2.1. Antibody-Based Therapies

The activation of T-cells is controlled by various costimulatory pathways which could be positive or negative. For example, signaling through CTLA4 induces an anergic state in naïve T-cells, and therefore Abatacept, a fusion protein composed of the Fc portion of human IgG1 fused to the extracellular domain of the CTLA4, is used for treatment of rheumatoid arthritis [63]. In a recent clinical trial, abatacept has demonstrated potential against T1D by delaying C-peptide exhaustion in T1D patients [41]. The clinical trial TrialNet is analyzing the benefits of abatacept in the delay of early T1D onset (Clinical trial NCT01773707; www.clinicaltrials.gov).

Anti-CD3 monoclonal antibodies that target CD3/T-cell receptor (TCR) complex, blocking the union of CD3 with TCR and rendering an anergic state of the T-cells have also been tested in T1D patients. Teplizumab and otelixizumab, two of the main clinically approved anti-CD3 antibodies, have demonstrated some efficacy in T1D patients [42]. Teplizumab treatment induces a delay in C-peptide decay in treated T1D patients. In this study AbATE, 2 week-teplizumab treatment resulted in C-peptide preservation [17] [Clinical trial NCT01030861; www.clinicaltrials.gov]. The main results from this clinical trial are expected to be released at the end of 2019.

Among the selective ablation of T effector cells, the elimination of memory T-cells would also be necessary in order to obtain long-lasting results. This could be achieved by inhibiting CD2 signaling. The anti-CD2 fusion protein Alefacept efficiently blocks T-cell activation inducing apoptosis of both memory and effector T lymphocytes. Unfortunately, only a modest trend towards preserving C-peptide levels was achieved when this hypothesis was tested during the clinical trial T1DAL, which included patients at late stage of T1D [18]. 

Lastly, anti-thymocyte globulin (ATG) have been described to be able to deplete activated T-cells. ATG used in low doses in combination with granulocyte colony-stimulating factor (G-CSF) showed that it is safe and can induce protection of beta cell mass [43]. Later clinical trials demonstrated that G-CSF by itself did not provide any additional benefits [64].

### 2.2. Proinflammatory Cytokine-Based Treatments

The role of inflammation and proinflammatory cytokines have been long known to have a role in T1D development [65]. Inhibition of expression of those molecules can induce important changes in pancreatic beta cells [44]. Such strategy was clinically used for treatment of other autoimmune diseases [65].

Interleukin (IL)-1α and IL-1β are important immunomodulators expressed by monocytes that can induce a toxicity on beta cells [45]. Anti-IL-1 administration for rheumatoid arthritis has been proven to be well tolerated in patients [46]. IL-1 is also involved in T1D progression by activating T helper cells, and improving the number of circulating memory T-cells [47]. A clinical trial performed on T1D patients suggested that IL-1 inhibition could induce a preservation of pancreatic beta cells [66].

Another cytokine that plays an important role as an intermediary molecule in autoimmune diseases is tumor necrosis factor α (TNF-α). Therefore, the blockade of TNF-α has been tested as a treatment of autoimmunity. Regarding T1D, patients that were treated with Etanercept (recombinant TNF-α receptor–IgG fusion protein) had improved preservation of beta cell mass (assessed by the C-peptide levels) and decreased glycated hemoglobin levels [48].

The ability of nicotinamide alone or in combination with vitamin E to preserve functionality of remaining beta cells has also been tested. Both treatments proved to be effective in retaining the basal secretion of C-peptide [67].

The IL-12/23 cytokine pathway, which is involved in the induction of inflammatory cytokines and pathogenic T-cell activation, was also considered as a potential therapy for T1D therapy. The application of Ustekinumab (IL-12/23 blocking molecule) has been tested in patients with T1D (UST1D clinical trial) (Clinical trial NCT02117765; www.clinicaltrials.gov).

Overexpression of IL-6 was noticed in a subset of T1D patients [68]. As a result, anti-IL-6 therapy, which is also tested in managing arthritis and systemic juvenile idiopathic arthritis [69], was initiated. Currently, the clinical trial EXTEND (Clinical trial NCT02293837; www.clinicaltrials.gov) is examining whether the blockade of IL-6 signaling (tocilizumab, an anti-IL-6 receptor antibody)) can induce a protection of beta cell function in T1D patients (ages 6 to 17 years) is ongoing.

Taking all the data together, cytokine inhibition is emerging as a viable supplementary approach in order to achieve durable therapeutic efficacy of T1D treatment.

### 2.3. T_reg_-Mediated Strategies

T_regs_ have also been involved in the pathophysiology of T1D [70]. Bluestone and colleagues examined the role of expanded autologous polyclonal T_regs_ in the treatment of T1D patients [70]. In this clinical trial, autologous T_reg_ infusions were safe, but did not modify the course of the disease. Other clinical trials have also evaluated the effects of low doses of IL-2 on T_reg_ activity [71]. Although IL-2 was able to increase the total number of T_reg_, this did not result in better glycemic control.

Intriguingly, recently published data has shown that blocking extracellular deposits of the polysaccharide hyaluronan (HA) (frequently observed in T1D patients [72]) reduced diabetes in two different mouse models by significantly enhancing the percentage of T_reg_ in pancreatic islets and preventing further β cell destruction [34]. 

### 2.4. Removal of Autoreactive T-cells

Targeted depletion of autoreactive T-cells in T1D patients is an approach with great potential, as it aims to eliminate effector T-cells responsible for the destruction of pancreatic beta cells. Treatment of NOD mice with anti-CD3 antibodies was shown to induce anergy in T-cells [49]. Additionally, elevated high counts of T_reg_ cells were observed in patients administered with anti-CD3. A clinical trial performed with T1D patients showed reduced insulin requirements after treatment with the anti-CD3 antibody [42,73]. No severe adverse events were observed, and even mild sides effects were rarely reported. These results suggest that anti-CD3 antibody treatment can be considered as a potential treatment for T1D [49].

### 2.5. B-Cell-Targeting Therapies

Since, B cells were implicated to take part in beta cell destruction through autoantibodies production, targeting of B cells in T1D settings has also been studied. The elimination of B cells in NOD mice prevented the accumulation of auto-antibodies, thus averting diabetes onset [74]. T1D patients treated with anti-CD20 antibodies showed higher C-peptide levels and lower insulin dependency when compared to the placebo group. However, this strategy does not seem to completely prevent C-peptide decay [50,75]. 

## 3. Antigen-Dependent Immunotherapy

In contrast to antigen-independent strategies, autoantigen-targeting treatments of T1D could modulate specifically T1D-related autoimmunity while preserving the normal immune homeostasis. The main objectives of antigen-specific therapies is to induce tolerance of autoreactive T effector cells and expansion of autoantigen-specific T_reg_ cells [38,39].

### 3.1. Beta Cell-Autoantigen Vaccination

The exposure of specific antigens to naïve T-cells could induce immune tolerance to that antigen. According to current knowledge of T1D progression, we can hypothesize that antigens derived from beta cells that are applied in a non-inflammatory context might modulate autoreactive T-cells, resulting in beta cell preservation [76]. This paradigm has led to developing novel vaccination strategies to achieve the induction of T-cell tolerance against specific autoantigens. The well-known T cell epitopes against insulin and glutamic acid decarboxylase (GAD) have been extensively studied [51], demonstrating that C19-A2 proinsulin peptide could modulate autoreactive CD4^+^ T-cells in patients with specific class II allele [77]. The administration of this peptide in recently diagnosed T1D patients resulted in the exhibition of higher C-peptide levels without symptoms of systemic or local hypersensibility [78].

Additionally, another T1D autoantigen, GAD65, was targeted in NOD mice in order to reduce the number of GAD65-specific T effector cells [79]. Normoglycemia was achieved in 70% of NOD mice, and in 80% of them normoglycemia persisted in long-term post-antigen administration. 

Despite the successful results observed with vaccination strategies in NOD mice, the dissimilarities in autoantigens between human and mice and the heterogeneity of T1D in humans makes this strategy not very suitable for clinical application [52].

### 3.2. Specific T-Cell Strategies 

The dysfunctional imbalance of T_reg_ to T effector cells is an important factor determining the onset of T1D [80]. CD8^+^ T-cell activation is a process mediated by the presentation of specific epitopes from professional antigen-presenting cells (APCs) as DCs appear to be the principal APCs for the CD8^+^ T-cell [81]. The process depends on CD4^+^ T-cells’ interaction that induce the activation of specific subsets of CD8^+^ T-cells which in turn is responsible for initiating islets’ beta cell destruction [53]. The process of achieving self-tolerant T effector cells could be through use of either the whole antigen or specific peptides. However, success in tolerization of T effector cells depends on different factors, especially the identification of the autoantigen that drives this process. In order to prevent beta cell destruction, the most relevant T effector clones have to be deleted. 

### 3.3. Specific B-Cell Strategies

The strategy based on the abrogation of non-specific B cells has not been very effective. However, inhibition of specific autoantigen B cells by depletion of insulin-reactive B cells, is a promising alternative [82]. Insulin-specific B cells elude the immune control in NOD mice responding to insulin by increasing the expression of costimulatory molecules during the crosspriming of effector T-cells.

## 4. Beta Cell Therapies

### 4.1. Replacement Therapies: Edmonton Protocol

The Edmonton protocol has shown the value of islet transplantation in addressing insulin regulation in T1D patients [7]. According to this protocol, pancreatic islets obtained from cadaveric donors are infused into immunosuppressed T1D patients. 

Trials conducted before 1990 using single islet infusions were partially successful, as they resulted in lower insulin needs and higher C-peptide levels; however, no additional steps to increase the net islet mass of the transplant had been taken in any of those trials [54]. 

Islet transplantation protocols became a promising therapy for type 1 diabetes thanks to the introduction of the Edmonton Protocol in 2000. Today this method is the only therapy that can reach glycemic control without the administration of insulin [55]. Transplantation of pancreatic islets has several advantages over the transplantation of a complete pancreas, since it involves only a minor surgical procedure with low morbidity and mortality, and a significantly lower cost. The main advantage of islet transplantation protocols over conventional insulin therapy is that transplanted islets are more efficient in maintaining normal blood glucose levels without producing excess insulin that could lead to episodes of hypoglycemia.

Modifications of the Edmonton Protocol based on a new immunosuppression regimen have prevented the use of corticosteroids, allowing the application of a unique combination therapy based on anti-interleukin-2 receptor antibodies along with the immunosuppressant drugs sirolimus and tacrolimus. The main advantage of this combination treatment is low beta cells toxicity. Islet transplantation has shown some success regarding insulin independence both in the short and long term [55,83] as much of the variability in the results obtained with the Edmonton Protocol is associated with factors related to both the organ donor and the recipient.

Although the benefits of the islet transplantation protocol are unquestionable, among the concerns for standardization of this strategy are the large number of islets that have to be transplanted and the adverse effects derived from the immunosuppression regimen. The first problem could be addressed by using stem cells that, under the adequate differentiation protocol, are able to differentiate into glucose sensitive insulin-producing cells (see Section 5.3).

### 4.2. Beta-Cell Regeneration Strategies

Gastrin and GLP-1 have a synergistic effect in inducing the regeneration and differentiation of beta cells [56,57]. In the NOD mouse model, the addition of both molecules resulted in increasing of beta-cell mass [58]. In addition, the combination therapy with DPP-4 inhibitors, (to increase GLP-1 levels), and proton pump inhibitors (PPIs; to increase gastrin levels), increased C-peptide levels and insulin secretion, and restored the normoglycemia in NOD mice [56]. In humans, the study REPAIR-T1D analyzed the effect of one-year similar treatment using a combination of sitagliptin (DPP-4 inhibitor) plus lansoprazole (PPIs inhibitor) in T1D patients [60]. However, no differences in C-peptide levels were observed between treated vs. placebo groups [60]. The authors claim that the increase in gastrin concentrations and GLP-1 were low, resulting in non-efficient treatment. Further clinical trials will be required in order to determine the role of gastrin and GLP-1 combination therapy.

## 5. Stem Cell Therapy Strategies

### 5.1. Tolerogenic DCs

Although various cell types have been studied as potential targets for T1D treatment, dendritic cells attracted special interest. However, clinical trials in which T1D patients received autologous DCs showed limited results. In these clinical trials, DCs were infused via abdominal intradermal injections every 2 weeks [61]. Although the treatment was well tolerated, no significant differences on glycaemia were observed.

Previous studies demonstrated, that dendritic cells, alone or via different effector cells, such as T_regs_ and B-regulatory cells (B_reg_), could play an important role in the activation status of autoreactive CD8^+^ cytotoxic T-cells (CTL) as well as influence the balance between T-helper cells (Th1 and Th2) and effector cell populations [59]. Tolerogenic DC (tDCs) populations have been used in different clinical trials for treatment of autoimmune diseases, including T1D [61,84]. The results of those studies suggested that tDCs remain at the administration site promoting the generation of a lymphoid stroma tissue which in turn allows the increase of FoxP3^+^ T_regs_ [85]. 

The synergistic inter-relationship of tDCs and T_regs_ allows them to generate a very powerful tolerogenic state. Co-administration of tDC and T_regs_, would allow stabilization of Foxp3 expression and would elevate the levels of IL-10, TGF-β, and retinoic acid by tDCs [86,87]. The tolerogenic state of the tDC would be increased via cell–cell interactions or through paracrine mechanisms. This combination strategy may change the paradigm of how autoimmune diseases are being treated, addressing the disproportion of the immune effectors generated during the disease-onset.

### 5.2. Hematopoietic Stem Cells (HSC)

Although immune dysfunctions linked to T1D are complex, Voltarelli and colleagues published an innovative research, where newly diagnosed T1D patients enrolled in a phase 1/2 clinical trial received immunosuppression treatment together with the infusion of autologous HSCs. The results obtained were promising; almost all patients did not require insulin injections for 6 months as their C-peptide levels stayed stable and the anti-GAD auto-antibodies levels were diminished [88]. 

In two recent prospective non-randomized trials, most patients showed no need for insulin administration after HSC transplantation [89,90]. The results of those studies showed that even 4 years post-transplantation, the C-peptide levels were still significantly higher than pre-transplant ones [89]. 

Recently, the results from a study using autologous non-myeloablative HSC transplantation were published [62]. Fifty-nine percent of the patients included in this clinical trial did not require insulin administration while 32% remained insulin-independent for at least 4 years [62].

Most of the patients included in the autologous HSC-transplantation clinical trials presented limited side effects. Only one clinical trial declared a patient death due to *Pseudomonas aeruginosa* sepsis [89].

Although the adverse effects related to immunosuppression protocol limit this alternative treatment, the administration of autologous HSC remains an exciting way forward in the task to find a cure for T1D.

### 5.3. Mesenchymal Stem Cells

Mesenchymal stem cells (MSCs) are stromal stem cells that play important roles in tissue repair and regeneration [91]. MSCs express specific antigen biomarkers (MHC I, CD90, CD105, and CD73) that enable their identification by flow cytometry techniques. MSCs have proven to be very promising in regenerative medicine thanks to their ability to give rise to different cell types, such as adipocytes, chondrocytes, and osteoblasts, making it possible to replace damaged tissues. [92]. In addition, MSC can be recruited from other injured tissues, such as ischemic heart or pancreas [92,93]. For this reason, MSCs are representing a new approach that will help the promotion of the integration of stem cell transplants in regenerative medicine protocols [94].

MSCs have been used to treat T1D patients and showed promising results in maintaining blood C-peptide levels [95]. However, no differences were observed for insulin requirements when compared with the non-treated group during the study.

The biological properties of MSCs regarding their potential to control aberrant immune response were demonstrated in NOD mouse model [96,97]. In Uppsala University Hospital’s sponsored clinical trial, in which T1D patients were transplanted with autologous MSCs, treated patients exhibited a better maintenance of C-peptide levels [96]. Umbilical cord blood MSCs (UC-MSCs) were also tested in combination with autologous mononuclear cells derived from bone marrow (aBM-MNC) in another clinical trial. The results of this study showed that the infusion of aBM-MNC induces a 30% reduction of insulin requirements [98]. Nowadays, many trials are trying to test the use of MSCs from different sources for the treatment of T1D, including the use of allogeneic MSCs derived from adipose tissue (NCT02940418 and NCT02138331).

To date, the use of immunoregulatory MSCs is a very promising topic in the T1D stem cells field. The combination of MSCs with other immunotherapies would offer a novel strategy for the treatment of T1D patients.

## 6. Novel Strategies

### 6.1. CAR-T-Cell Therapy

#### 6.1.1. Introduction

In the recent years, an immunotherapy using engineered T-cells expressing chimeric antigen receptors (CARs) specific against CD19 emerged as a major breakthrough in cancer therapy of CD19+ B-cell leukemia [99]. CARs are complex molecules composed of several components, the most common being: (1) An antigen-specific recognition domain, usually a single chain variable region (scFv) from a monoclonal antibody; (2) a hinge region, based on the Fc portion of human immunoglobulin (IgG1 or IgG4), or originating from the hinge domains of CD8a or CD28; (3) a transmembrane domain; and (4) an intracellular tyrosine-based signaling domain [100]. The signaling domain is the engine of the receptor. Its most common component is the intracellular portion of CD3ζ, which is the main signaling chain of CD3 T-cell receptor (TCR) complex. The biggest advantage of CAR-T-cells is that the receptor’s interaction with its antigen is independent from major histocompatibility complex (MHC) but it still activates the same TCR’s and costimulatory intracellular signaling cascades necessary for T cell activation and expansion. 

#### 6.1.2. CAR-T-Cells and T1D

Based on the studies with CARs in cancer and increased interest of T_regs_ as a potential tool for T1D therapy (see Section 2.3). It is only logical to hypothesize that armoring T_regs_ with β cell-specific CARs would improve T_regs_’ migration into the pancreas and pancreatic lymph node, thus protecting islet cells from autoimmune destruction. A number of recent studies suggests that there is big potential for CAR-T_regs_ therapy in multiple autoimmune or allograft rejection model systems [101,102,103,104,105,106]. Fransson and colleagues described an interesting approach for CAR-T_regs_ use in the EAE mouse model [105]. In their study, CD4^+^ T-cells were engineered to express both a CAR specific against myelin oligodendrocyte glycoprotein (MOG_35-55_) and a murine Foxp3 gene to drive T_reg_ differentiation, separated by a 2A peptide sequence. Intranasal administration of CAR-T_regs_ resulted in a successful delivery to the CNS, an efficient suppression of the ongoing inflammation and complete recovery from disease symptoms. Other studies propose the use of CAR-T_regs_ in transplant rejection by generating HLA-A2-specific CAR-T_regs_ that were isolated from the host [102,104]. These HLA-A2-CAR-T_regs_ retained high expression of Foxp3, LAP, GARP, and CTLA-4, and maintained their suppression function *in vitro* without a significant cytolytic activity. Even though there is still necessity to confirm the stability of T_reg_ phenotype, purity, and long term survival after the transfer, this approach is very promising for treating and prevention of transplant rejection by inducing graft-specific tolerance. 

CAR-T_regs_ were also studied in Hemophilia A, where genetic mutations in F8 gene result in either reduced levels or altered functionality of the blood-clotting protein, Factor VIII (FVIII). In patients with severe hemophilia (no circulating FVIII can be detected), there is a high probability for developing adverse immune reactions to the exogenously administered FVIII protein. Remarkably, administering FVIII-specific human CAR-T_regs_ suppressed antibody production *in vitro* and *in vivo* in a mouse hemophilia A model. However since FVIII is a soluble protein, the mechanism of this suppression is not entirely clear [101,107]. 

Hansen’s group study was an additional proof of concept that CAR-T_regs_ are a prospective therapy strategy for multiple autoimmune conditions [106]. The authors generated CAR against carcinoembryonic antigen (CEA), a glycoprotein presented on lung adenoepithelia, and then adoptively transferred T_regs_ expressing this construct in an experimentally induced allergic asthma mouse model. The CAR-T_regs_ accumulated in the lungs and nearby lymph nodes, reducing airway hyper-reactivity, inflammation, mucus production, and eosinophilia.

#### 6.1.3. Challenges

Despite the great potential of CAR-T_regs_ therapies, there is still no clear strategy on how to use this exciting technology for the treatment of T1D. The biggest challenge is the lack of β cell-specific antibodies that can be harnessed to generate islet-protective CAR-T_regs_. One possible approach to overcome this problem is to use human islet-specific TCR gene transfer to polyclonal human T_regs_. A recent study where polyclonal T_regs_ were transduced with TCR chains derived from two human islet-specific CD4^+^ clones showed an improved antigen-specific suppression of these cells and increased potency when compared to polyclonal T_regs_ [108]. However, such islet-specific T_regs_ were less responsive to their cognate antigen in comparison to T-cells expressing virus-specific TCRs suggesting that further optimization and/or identifying better TCR clones is still needed. 

A new study demonstrated that insulin-specific CAR-T_regs_ were functional, suppressive and surviving *in vivo* even though they were not able to prevent spontaneous diabetes in NOD mice [109]. This is not a surprise considering the fact that insulin is a soluble antigen that is present throughout the body and its concentrations fluctuate. Moreover, such a strategy would not be very efficient in patients with T1D where endogenous insulin levels are very low and the daily insulin injections would disturb the normal insulin concentration gradient that might drive the insulin-specific CAR-T_regs_ into the pancreas. 

Therefore, the discovery and study of new β cell-specific molecules that could provide proper targeting of CAR-T_reg_s is needed. While there are some promising molecules such as DPP6 [110]), FXYD2γa [111], and NTPDase3 [112], all of those would require additional studies confirming their specificity, as well as isolating appropriate monoclonal antibodies that would recognize human β cells *in vivo* before developing a CAR construct for T1D therapy.

#### 6.1.4. Summary

In summary, despite the advances in the field of CAR-T_regs_ therapies and their great potential to be applied for autoimmune disorders, there is still a lack of an efficient system as well as of appropriate surface β cell-specific markers that would allow the generation of effective auto Ag-specific T_regs_ that could be used for cell-based therapies in T1D.

### 6.2. Microbiota Modulation

#### 6.2.1. Introduction

The microbiota refers to a complex ecosystem of bacteria and viruses, among other microorganisms that inhabits our body, especially the digestive tract. This community greatly exceeds the amount of eukaryotic cells that form the human body and their collective genome, named microbiome, is considerably larger than the human genome. On account of the mutualistic relationship between the host and its gut microbes, the imbalance of the latter, which is termed dysbiosis [113], could spoil gut microbiota (GM) physiological properties leading to harmful effects to the human host [114]. 

Among the GM properties, there are important metabolic benefits such as improving the digestive functions. Bacteria allow the complete digestion of some food nutrients such as fibers which otherwise cannot be metabolized by eukaryotic cells [115], and participate in the synthesis of some micronutrients [116]. Importantly, some relevant functions have been described for metabolic end products of microbial fermentation. For example, during metabolism of fibers, short-chain fatty acids (SCFA) such as butyrate, propionate, and acetate are produced [117]. The former is of great importance and acts as an energy source for colonic epithelial cells, thus contributing to the proper barrier function [118,119]. Besides its nutritional impact, the current evidence supports the fundamental role of the GM in the host defense. The intestine works as a boundary that separates the inner and the outer environment and the coexistence of microbial and somatic cells is highly mediated by the epithelial cells (EC). This complex system was well illustrated by Vaishnava et al. who emphasized the interplay between EC and gut microbes and its significance for their proper coexistence [120].

The mechanisms underlying the cross-talk between the gut microbial community and the immune system (IS) are well stablished and it has now became clear the relevance of such interplay in the harmonious balance between the host and its microbiota [114]. The mucosal IS, which is distributed among the different levels of the mucosa layer, has to procure the right equilibrium between tolerance and reactivity, and T-cells are decisive for such balance [121]. Distinct T-cell sub-populations dominate in different gut locations, conditioning the immune activation through complex signaling pathways (28). Because of the impact of both commensal and pathogenic bacteria on the maturation of the IS, the study of the microbiota and gut integrity may clarify the field. 

Today there is clear evidence of the relevance of an adequate development of the microbiota and immunity for the host wellness. Data from experimental studies on *in vivo* models have provided valuable knowledge. Findings from germ-free animal studies revealed important phenotypic and functional characteristics mediated by the intestinal microbes, and emphasized the importance of the microorganisms in the correct development of the human body structures [122]. Certainly, studies on the transfer of microbiota from humans or animal models to animals with known microbiota (gnotobiotic models) are prevalent and they demonstrate that some phenotypic characteristics are dependent on the microbiota [123].

#### 6.2.2. Microbiota and T1D

Vaarala et al. elegantly described the three main elements that may explain the connections between an altered intestinal track and T1D [124]. This triad includes a compromised gut permeability, immune dysregulation, and a dysbiotic microbial ecosystem. Additionally to the defective barrier function and intestinal environment, confirmed in later studies in T1D subjects [125], the microbes play a key role also in the development of T1D. For example, the number of anti-islet cell autoantibodies has been shown to correlate with some bacteria genera, suggesting that alterations in the microbiota composition may precede the pathology. Indeed some degree of gut dysbiosis has been observed in prediabetic subjects prior to T1D onset [126]. 

There is accumulating evidence of the role of GM in diabetic pathology. In fact, a divergent profile of intestinal bacteria has been reported in T1D individuals in comparison to non-diabetic subjects. A case-control study with a total sample of eight children, four cases and four controls, revealed that T1D patients possess distinctly different gut microbiota, compared to healthy subjects, characterized by an increased Bacteroidetes/Firmicutes ratio [127]. The same finding was reported in a later study on Chinese T1D subjects [128]. Giongo et al. emphasized that changes at phyla levels were essentially a result of shifts in specific genera; *Clostridiales* and *Bacteroides* in Firmicutes and Bacteroidetes, respectively. They also found a list of bacteria genera predominant in the diabetic and control children [127]. In a related publication, the same research group provided further findings regarding the GM composition in the same sample [129]. It should be noted the increased abundance of advantageous bacteria such as butyrate producing bacteria (BPBs) and mucin-degrading bacteria in healthy controls [129]. The former bacteria group is known to enhance the barrier function through the maintenance of the mucus layer as mentioned above. The later contributes with a better permeability by means of mucin production, aiding in a steady mucus layer as well as gut integrity [130]. A compromised presence of BPBs and the consequent decay of the barrier function is thought to be a primary trigger of pro-diabetic intestinal profile. *Akkermansia* genus, specifically *A. muciniphila* is probably, along with the *Faecalibacterium* genus, the most studied BPB. This taxa is specifically associated with the mucus layer by participating in its regulation through mucin degradation and human studies showed an association of its depletion with compromised mucus integrity [131]. Besides its structural role, *A. muciniphila* may have an effect in the defense response and *in vivo* studies demonstrated a function in the immune regulation by the activation of immune cells [132]. Indeed, children with T1D presented an under-abundance of *A. muciniphila* compared to controls [130], in concordance with the compromised microbial butyrate production observed in the NOD mice [133]. The restoration of *A. muciniphila* representation in type 2 diabetic mice also triggered important phenotypic features along with improvements in the barrier function [132]. These findings suggested that *A. muciniphila* could be a key player in the prevention and management of aberrant microbiota associated with T1D and related autoimmune diseases.

Likewise, microbial diversity appears to be impaired in T1D. A study using samples from eight Finish children in which four case children later developed T1D and the other four were controls, revealed that the case children’s samples had an unsatisfactory development in GM diversity, which did not become as complex as controls’ and was more heterogeneous among cases [127]. The same finding was reported by Kostic et al. [134]. Giongo et al. emphasized the importance of a compromised phylogenetic diversity in the risk of developing autoimmune diabetes and set the basis of potential screening criteria. Additionally, some functional attributes of the microbiome has also been reviewed in relation to T1D. Brown’s team went further and detailed functional differences between controls and cases [129], revealing a greater taxonomic complexity in the control group. Conversely, a reduced metabolic capacity found in cases was associated with lower microbial diversity and predominance of unwanted bacteria taxa such as those matched to a pro-inflammatory state [127,134].

Long cohort studies and randomized controlled trials such as FINDIA (Finnish Dietary Intervention Trial for the Prevention of Type 1 Diabetes), BABYDIET (in German infants), TRIGR (*Trial to Reduce IDDM in the Genetically at Risk*) and TEDDY among others, offer valuable information regarding the natural history of T1D and the role of GM (reviewed in [123]). Within the findings, the effect of geographical location on intestinal microbiota has received considerable attention. Other *in vivo* studies have contributed with valuable knowledge as well. For instance, Kriegel and colleagues correlated the abundance of intestinal segmented filamentous bacteria (SFBs) with the development and progression of diabetes in NOD mice [135]. Although a protective role for SFBs could not be presumed, they concluded that SFBs somehow attenuates the progression of T1D and promotes a boost in some T helper cell sub-populations. SFBs were initially considered latent but the current evidence clues that they have a role in mucosal immunity and immune response. 

The features and characteristics of a pathogenic T1D-prone microbiome seems to precede the disease, which offers a possibility to anticipate and prevent or delay T1D onset [118,123,134]. Therefore, the GM could be used as a potential marker for disease progression. For instance, some specific bacteria taxa, such as the Ruminococcaceae family, have proven to have an inverse relationship with the levels of serum hemoglobin A1c [128], a widely used biomarker for the evaluation of diabetes progression. 

A large number of experimental and observational studies demonstrated the efficiency of both probiotic and prebiotics, as well as synbiotics and fermented products, in conferring benefits on the host [136]. Thought probiotic efficiency is specie-dependent, and some methodological and technical issues such as the dose or the capacity to colonize the gastrointestinal track may limit their efficiency [137], this approach seems promising for T1D. Along with the aforementioned dietary modulations, fecal transplants also offer a possibility of changing host’s microbiota. The fecal microbiota transplantations (FMTs) were initially used in experimental studies [138] but has proven to be effective in the management of some intestinal pathologies [139] and its use in T1D has been discussed [140]. Despite the controversy about its use, FMTs may be a useful tool for immunomodulation and seems to be a promising approach for the GM modulation.

Some novel publications discuss the relevance of the aforementioned products for T1D management [141,142]. Interestingly, studies in mice models [143] and humans [144] reported beneficial outcomes after intervention with potentially beneficial bacteria. For instance, the administration of the probiotic *A. muciniphila* showed an improved insulin sensitivity and glucose homeostasis, healthier lipid profile, and a pro-inflammatory tone among others changes. Interventions that aimed to promote *A. muciniphila* abundance through a prebiotic effect [144,145] offered positive effects as well.

#### 6.2.3. Summary

The above reported findings provide convincing evidence that the GM should not be dismissed on the management of T1D. Available information, especially prospective human studies, seems to suggest a main role of the GM in the risk and development of autoimmune disorders. Efforts to identify specific targets in the GM would help to improve the effectiveness of these novel approaches and provide diabetic patients with alternative medical treatments.

### 6.3. JAK Pathway Inhibition

#### 6.3.1. Introduction

The mammalian Janus kinase (JAK) family contains three JAKs (JAK1, 2, 3) and tyrosine kinase 2 (TYK2), which selectively bind different receptor chains [146]. Upon binding of ligand to its cognate receptor, associated JAKs become activated and undergo phosphorylation, which creates docking sites for the SH2 domain of the cytoplasmic transcription factors termed signal transducers and activators of transcription (STATs). The human STAT family contains seven STATs: STAT1, STAT2, STAT3, STAT4, STAT5A, STAT5B, and STAT6. Following phosphorylation, STATs are translocated to the nucleus, dimerize, and bind to specific DNA sequences to regulate gene transcription [147]. The JAK-STAT pathway plays a pivotal role for the downstream signaling of inflammatory cytokines, such as IFNs, ILs, and growth factors [148].

#### 6.3.2. JAKs and T1D

A type I IFN signature precedes the detection of autoantibodies in children genetically at risk for T1D [149] and IFNα is expressed in human islets from type 1 diabetic patients [150,151]. MHC class I overexpression is induced by IFNα [152] and IFNγ [153] in human islets from T1D patients and IFNα also induces β cell endoplasmic reticulum stress and chemokine production [154]. 

Receptor engagement by IFNα triggers JAK1-TYK2 heterodimer signaling (Figure 2). TYK2 has been associated with several autoimmune diseases including rheumatoid arthritis and T1D [155,156]. Six TYK2 single nucleotide polymorphisms (SNPs) (rs34536443, rs2304256, rs280523, rs280519, rs12720270, and rs12720356) have been explored in relation to autoimmunity. Crucially, the SNP rs2304256 causes a missense mutation in TYK2, and has been associated with protection against T1D [155]. 

Downstream IFNα/IFNγ signaling is STAT1 dependent (Figure 1), and STAT1 is overexpressed in T1D islets and strongly correlates with HLA class I expression in β cells [153]. 

IFNγ is also involved in the expression of the CXCL10, which seems to be activated in islets from both T1D patients [157] and non-obese diabetic (NOD) mice [157,158]. CXCL10 promotes pathogenic T-cell infiltration into the pancreatic islets leading to β cell apoptosis and its neutralization prevents diabetes in NOD mice [159]. A lack of IFN-γ delays the progress of autoimmune diabetes in NOD mice [160]. 

Recent evidence further supports the rationale that IFN-driven JAK-STAT pathway activation significantly contributes to T1D pathogenesis. Patients with STAT3 gain-of-function germline mutations are susceptible to T1D with the median age of onset being 8 weeks. Furthermore, approximately 15% of patients treated with immune checkpoint inhibitors develop endocrine autoimmunity [161], including pancreatic β cell targeting [162], leading to T1D [163]. Consistent with these observations, inhibition of PD-1-PDL1 signaling accelerates diabetes in NOD mice [164]. 

Prior treatment of *in vitro* human islets with ruxolitinib (JAK1/2) significantly reduced IFNα mediated inflammatory and ER stress markers [165]. Moreover, treatment of NOD mice with a JAK1/JAK2 inhibitor (AZD1480) blocked MHC class I upregulation on β cells and reversed autoimmune insulitis by reducing immune cell infiltration into islets in newly diagnosed animals [166]. 

Finally, pancreas-specific genetic knockout studies revealed an essential role for STAT3 in islet architecture, but it is dispensable for the function of mature islet [167,168]. In contrast, STAT5 is only important for age-dependent glucose intolerance [169]. These studies suggest that β cell function is minimally impacted by JAK-STAT pathway inhibition.

#### 6.3.3. Summary

Taken together, IFN driven T1D pathogenesis can be potentially downregulated by inhibiting the downstream JAK-STAT pathway.

## 7. Concluding Remarks and Outlook

Diabetes is a complex disease that originates from dysfunction and destruction of beta cells as a result of a pathogenic response that involves both the adaptive and innate immune system [170,171]. During T1D development, T-cells seem to play a crucial role for destruction of beta cells [172]. Therefore, T-cells have been target of most immunotherapy strategies, dues to the main hypothesis that beta cells could survive by suppressing the pathogenic reactivity of specific T-cells. Although these strategies have demonstrated to be effective, unfortunately, the efficacy was short-lived. On the other hand, immunotherapy protocols based on specific antigens, such as vaccination with peptides derived from beta cells, should take into account the high degree of diversity in the response of specific T-cells against beta cells among individuals with T1D [173]. For this reason, the most effective approach should contemplate the combination of different strategies in order to allow the elimination of islet-infiltrating T effector cells through different mechanism. In this sense, new strategies with the objective of improving glycemic control are constantly investigated with the goal to address the long-term insulin dependence that leads to a poor quality of life.

In addition to immune interventions, other ongoing studies are investigating ways to restore insulin secretion using different approaches. It is important to note that, due to the heterogeneity of T1D, the future of T1D treatment strategies most probably would be in direction of a more personalized approach.

## Figures and Tables

**Figure 1 ijms-20-04789-f001:**
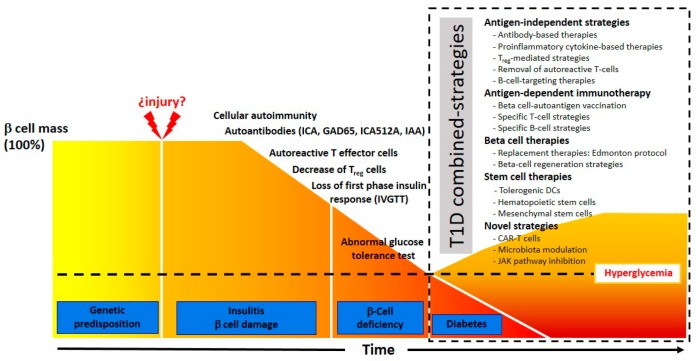
Progression of T1D and combined-strategies for T1D treatment.

**Figure 2 ijms-20-04789-f002:**
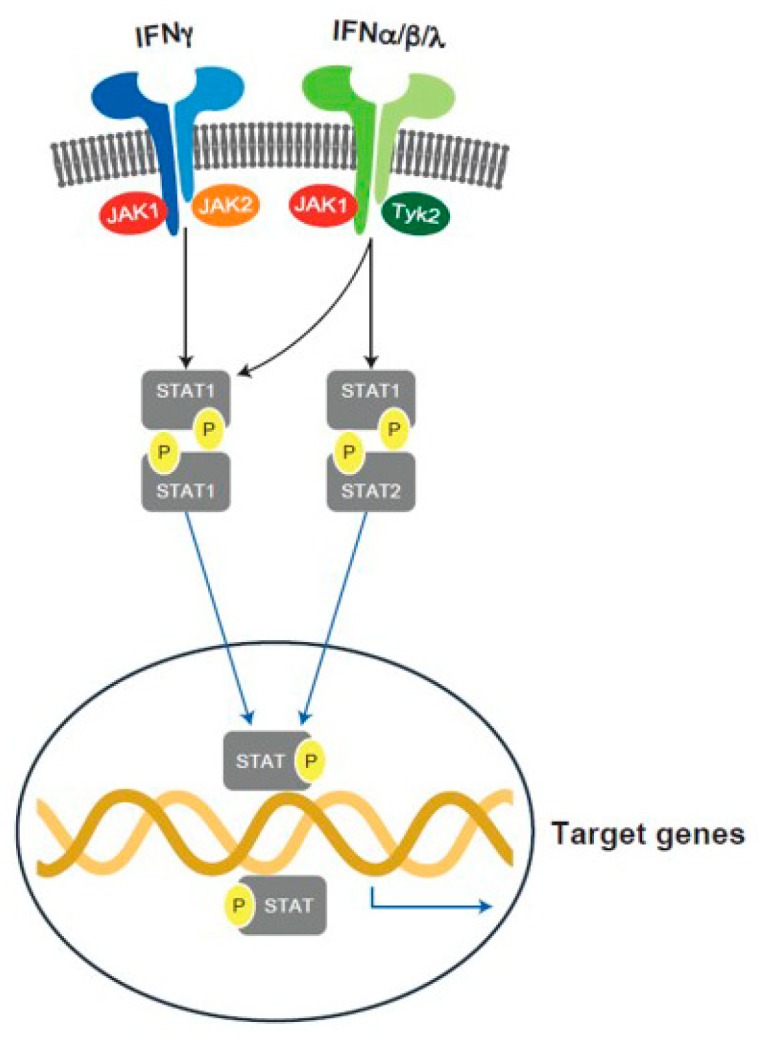
JAK1-TYK2 heterodimer signaling pathway.

**Table 1 ijms-20-04789-t001:** Autoimmune diabetes developed by NOD mouse compared to human T1D.

	NOD	Human
Age at onset	> 10 weeks	>6 months–late adolescence
Genetic susceptibility	MHC most important	HLA most important
Autoantigens	Insulin, GAD, IA-2, IA-2b, ZnT8, IGRP, Crhomogranin A	Insulin, GAD, IA-2, IA-2b, ZnT8, IGRP, IAPP, HSP60, Carboxypetidase H
Insulitis	DCs, Macrophages, B cells, NK cells, CD4 & CD8 T cells	DCs, Macrophages, B cells, NK cells, CD4 & CD8 T cells
Ketoacidosis	Mild	Severe
Gender effect	Females predominantly affected	Males and females almost equally affected

**Table 2 ijms-20-04789-t002:** Strategies for the treatment of T1D.

Antigen-Independent Strategies		References
	Antibody-based therapies	
	Anti-CTLA-4	Clinical trial NCT01773707
	Anti-CD3	Clinical trial NCT01030861
	Anti-CD2	[18]
	Anti-thymocyte globulin (ATG)	[35]
	Proinflammatory citokine-based therapies	
	IL-1a/IL-1b	[41]
	TNF	[42]
	Nicotinamide	[43]
	IL-12/23	Clinical trial NCT02117765
	IL-6	Clinical trial NCT02293837
	T_reg_-mediated strategies	
	Treg suppression	[44,45]
	Removal of autoreactive T-cells	
	Anti-CD3	[46,47]
	B-cell-targeting therapies	
	Anti-CD2	[48]
**Antigen-dependent immunotherapy**		
	Beta cell-autoantigen vaccination	
	GAD	[49]
	Specific T-cell strategies	
	Tolerized T effector cells	[50]
	Specific B-cell strategies	
	Depleting insulin-reactive B cells	[51]
**Beta cell therapies**		
	Replacement therapies	
	Edmonton protocol	[7]
	Beta-cell regeneration strategies	
	Gastrin + GLP-1	[52,53]
**Stem cell therapy strategies**		
	Tolerogenic DCs (tDCs)	
	Autologous tDCs	[54,55]
	Combination tDC + Tregs	[56,57]
	Hematopoietic stem cells (HSC)	
	Autologous myeloablative HSC transplantation	[58]
	Autologous non-myeloablative HSC transplantation	[59]
	Mesenchymal stem cells (MSC)	
	Autologous MSCs	[60,61]
	Allogeneic adipose-derived MSCs	Clinical trial NCT02940418
	Umbilical cord blood MSCs (UC-MSCs)	[62]

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
