# Peer review of "New Insights into Immunotherapy Strategies for Treating Autoimmune Diabetes"

_ijms, 2019, doi:10.3390/ijms20194789_

Round 1

Reviewer 1 Report

The review by Cabello-Olmo et al. is a nice overview of autoimmune diabetes treatment strategies.

I only have minor requests and suggestions:

professional English language editor should be consulted

line 15 "T1D has been successfully treated" is a bold statement

line 24 pretend to analyze? please clarify content of lines 30-33 

line 34-38 different spacing and size please be consistent with T cells or T-cells throughout the whole manuscript

line 80 please explain c-peptide

line 84, other T1D pancreases have reported?

1.3 animal models of T1D add a paragraph about the DORmO mouse (DO11.10xRIPmOVA)

line 111-112 list a few immune interventions in NOD mice

1.4 add references to the text

2.3 T-reg mediated strategies add studies about 4-MU (Nagy et al. JCI 2015)

6 Novel strategies, please break the sections down into smaller subsections, otherwise too long too read

Author Response

Author's Reply to the Review Report (Reviewer #1)

We appreciate the positive review (“a nice overview of autoimmune diabetes treatment strategies”) and thoughtful critique of the manuscript. We have answered all of the reviewer comments, and therefore we believe the current version better represents the current status of T1D immunotherapy.

Minor comments

Professional English language editor should be consulted

We agree with the reviewer that the original manuscripts contained multiple grammar mistakes, but we believe all those mistakes have been corrected in the heavily edited new version.

line 15 "T1D has been successfully treated" is a bold statement

We agree with the reviewer and that sentence has been modified by the sentence “Currently, T1D is treated by lifelong administration of novel versions of insulin…”. [line 15]

pretend to analyze? please clarify content of lines 30-33

We agree with the reviewer that the sentence was inadequate and has been clarified in the submitted new version of the manuscript including the sentence: “In the present review we explore the current state of immunotherapy in T1D…” the clinical consequences of this disease. [line 24]

line 34-38 different spacing and size please be consistent with T cells or T-cells throughout the whole manuscript

Corrected.

line 80 please explain c-peptide

We thank the reviewer for this suggestion. It has been added to the text a brief definition of c-peptide. [lines 78-79]

line 84, other T1D pancreases have reported?

We agree with the reviewer that the sentence was unclear and has been clarified in the new version, including the sentence: “the lack of insulitis in some T1D cadaveric pancreata samples…”. [line 83]

1.3 animal models of T1D add a paragraph about the DORmO mouse (DO11.10xRIPmOVA)

We agree with the reviewer that this an important model, and we have now included a brief description of it. [lines 122-128]

line 111-112 list a few immune interventions in NOD mice

We have now included some examples as well as references. [lines 110-111]

1.4 add references to the text

Included.

10 2.3 T-reg mediated strategies add studies about 4-MU (Nagy et al. JCI 2015)

We again agree with the reviewer that this an interesting paper and have included it in the manuscript.

6 Novel strategies, please break the sections down into smaller subsections, otherwise too long to read

Agreed. We have now subdivided the sections into 3-4 subsections, making it easier for the reader to follow.

We hope that our response and the changes that have been incorporated will be acceptable. Thank you very much in advance for considering our revised manuscript.

Reviewer 2 Report

Authors meticulously described a meaningful review on immunotherapy for type 1 diabetes. The review article is well written and it is easy to understand overall. The article deserves to be published without revision in the journal.

Author Response

We appreciate the extremely positive review (“Authors meticulously described a meaningful review on immunotherapy for type 1 diabetes. The review article is well written and it is easy to understand overall. The article deserves to be published without revision in the journal.”)

This manuscript is a resubmission of an earlier submission. The following is a list of the peer review reports and author responses from that submission.